# OpenReview forum: "Energy-Oriented Alignment for Large Language Models"
_ICLR.cc/2025/Conference — ICLR 2025 Conference Withdrawn Submission_

### Official Review · Reviewer_Byie · 2024-10-17

**Soundness:** 3
**Presentation:** 2
**Contribution:** 2
**Rating:** 3
**Confidence:** 3

**Summary:**

This paper introduces energy-oriented malicious instructions, a new attack on LLMs that forces models to generate excessively long outputs, consuming significant resources and risking system overloads. The authors present the EnergyAlign dataset with eight types of instructions to exploit this vulnerability. Experiments on 5 proprietary and 24 open-source LLMs show that open-source models are more vulnerable. The paper also explores jailbreak techniques that bypass proprietary models' defenses. The authors proposes energy-oriented alignment, a defense mechanism that trains models to reject such energy-wasting inputs, emphasizing the need for further research into robust defenses.

**Strengths:**

1. The paper identifies a new threat by examining energy-oriented malicious instructions, an attack that targets computational resource exhaustion.
2. The creation of the EnergyAlign dataset provides a useful tool for assessing models' vulnerabilities to energy-based attacks.
3. The evaluation of both proprietary and open-source LLMs offers insights into their differing levels of resilience.
4. The proposal of energy-oriented alignment as a defense mechanism represents a step toward improving model robustness in real-world applications.

**Weaknesses:**

1.  While the title mentions "alignment," much of the paper focuses on the dataset's effectiveness and relevance, leading to a disconnect in the writing.
2. Some of the open-source models such as the Llama2 series utilized in the paper are outdated, which may affect the representativeness of the results.
3. The paper does not provide sufficient insights into why the dataset is effective, nor does it delve deeply into why these specific points were chosen for dataset construction.
4. The presentation of results is simplistic, lacking more complex analyses or visualizations to support the conclusions.
5. The paper does not clearly describe the specific training settings used during the alignment process, making it difficult for readers to replicate or fully understand this aspect.

**Questions:**

1. If LLaMA3-3.2 were used for new experiments, would the conclusion that open-source models are more vulnerable still hold?
2. Can more analysis be provided on the effectiveness of EnergyAlign, i.e., why the specific categories of malicious instructions were chosen? Are there theoretical insights to its effectiveness?

---

### Official Review · Reviewer_2od3 · 2024-11-01

**Soundness:** 3
**Presentation:** 3
**Contribution:** 2
**Rating:** 3
**Confidence:** 3

**Summary:**

This paper explores the concept of energy-oriented malicious inputs targeting large language models (LLMs). Similar to Denial-of-Service (DoS) attacks, this paper focused on inducing models to generate excessively long outputs that strain computational resources. The authors introduce EnergyAlign, a dataset specifically designed for this type of threat. They conduct comprehensive evaluations on five proprietary LLMs and 24 open-source LLMs.

**Strengths:**

1. This paper is well-written and well-structured. The concept is clearly presented and easy to understand.
2. The evaluation is thorough and thoughtfully executed. Enhancing the safety and efficiency of LLMs is an important topic, and the energy-oriented perspective is new.

**Weaknesses:**

The reviewers' concerns are primarily on the feasibility and legitimacy/utility of the proposed energy-oriented malicious input: whether these malicious inputs are truly meaningful and necessary in invading and exhausting the computational resources of service providers.

1. Service providers, such as OpenAI, assign token limits for each user, for instance, initial accounts often have restricted token consumption per minute. This measure should be sufficient to counter such attacks.

2. While it could be argued that attackers might create numerous accounts to carry out energy-draining activities, this scenario falls under the domain of DoS (Denial of Service) defense and is not directly related to LLMs.

3. Most importantly, if the goal is simply to occupy a service provider's energy, attackers could achieve this by just sending normal requests. As long as the provider is required to generate tokens, their resources will be consumed. Malicious inputs like repetitive or random content are unnecessary for this purpose.

Thus, In the reviewer’s view, the proposed concern is not practically feasible in real-world scenarios.

Another concern arises from the distinction between energy-oriented malicious input and LLM instruction-following capabilities. Table 1 outlines various scenarios categorized as malicious input, but many of these are fundamental to LLM instruction-following applications, such as nested IF/ELSE statements, which have proven useful in contexts like verbalized ML [1]. The designed cases show significant overlap with legitimate instruction-following functions. Therefore, the reviewer is concerned about the potential for excessive refusal of valid input.

Building on the previous concern, the authors did not demonstrate the utility of LLMs after applying EnergyAlign. Specifically, there is no information on whether EnergyAlign causes LLMs to become overly cautious, leading to refusal of legitimate tasks, or how fine-tuning impacts the performance of benign tasks. Additionally, regarding defense mechanisms, did the authors experiment with simple system prompts to mitigate these "malicious" inputs? How effective were these prompts?

Reference:

[1] Xiao, Tim Z., et al. "Verbalized Machine Learning: Revisiting Machine Learning with Language Models." arXiv preprint arXiv:2406.04344 (2024).

**Questions:**

See above.

---

### Official Review · Reviewer_5Pre · 2024-11-03

**Soundness:** 3
**Presentation:** 2
**Contribution:** 2
**Rating:** 5
**Confidence:** 4

**Summary:**

This paper proposes an attack on LLMs aimed at increasing computational load. The authors suggest that instructing LLMs to generate meaningless long texts can lead to significant resource consumption, effectively creating a DoS-like attack. To explore this, they introduced a dataset called EnergyAlign, which aims to induce LLMs to generate extended outputs, and combined it with certain jailbreak methods to enhance the attack.

**Strengths:**

s1: The topic is intriguing, and the proposed attack method could potentially be used for DDoS attacks against LLMs, making it practically relevant.

s2: The paper includes extensive experimental results, with evaluations conducted on both open-source and closed-source models.

s3: I particularly appreciate the quantitative analysis provided in Appendix A regarding the effect on input and output token quantities.

**Weaknesses:**

w1: The paper seems to lack a strong technical contribution. It primarily relies on manually designed prompts to induce LLMs to produce longer outputs, which are then combined with existing jailbreak methods. However, there may be more effective ways to achieve the stated motivation. For instance, adversarial optimization could be used to make the LLM recursively generate the same content, or minimizing stop tokens (e.g., <\s> in Llama2) could prevent the LLM from stopping.

w2: In the discussion of combining EnergyAlign with jailbreak methods, the authors mainly justify the use of LongGPT and Sample-aware prefix. Could you provide more reasoning regarding the use of these jailbreak techniques?

w3: Potential adaptive defense methods are not considered. For example, powerful models could have a system prompt stating "Determine if the user's input is intended to generate large amounts of meaningless content; if so, refuse to respond," or use robust training techniques.

w4: It seems challenging to distinguish between the proposed energy-oriented threats and legitimate requests in some situations. For instance, developers may need LLMs to generate test samples following specific patterns, which is similar to the "Random" type in EnergyAlign. Could the use of EnergyAlign for robust training potentially harm the model's utility?

w5: Compared to inference time, a more common scenario involves using LLM service provider APIs, which typically calculate costs based on token usage. The price difference between input and output tokens may not be significant. In this case, would using large amounts of random text as input be more effective? Additionally, this approach could be harder to defend against.

**Questions:**

q1: Does the energy analysis in Appendix A account for the baseline power consumption of GPUs? Since the computation duration may vary.

q2: There is a lack of specific implementation details for prefix injection and sample-aware prefix. Could the authors further explain how to get the LLM to respond with a fixed prefix? Was a GCG-like method used?

q3: Many LLM services currently use caching, storing common input/output pairs. Could this mitigate the effectiveness of the proposed attack?

---

### Official Review · Reviewer_G7N7 · 2024-11-05

**Soundness:** 2
**Presentation:** 3
**Contribution:** 2
**Rating:** 5
**Confidence:** 4

**Summary:**

The paper introduces EnergyAlign, a novel dataset that assesses the resilience of large language models (LLMs) against energy-oriented malicious instructions designed to induce high computational loads. Unlike malicious inputs focused on toxicity or misinformation, energy-oriented attacks exploit LLMs to produce lengthy responses, maximizing energy and resource consumption. This paper evaluates 5 proprietary and 24 open-source LLMs, revealing that while proprietary models exhibit some resistance, open-source models are highly vulnerable, with failure rates reaching 96.8%. The paper also examines the effectiveness of jailbreak techniques in circumventing proprietary LLM defences and finds existing mitigation strategies inadequate. By benchmarking current defences and proposing energy-focused alignment strategies, this paper highlights an urgent need for more robust defences to safeguard LLMs from these novel energy-oriented threats​.

**Strengths:**

- This paper introduces EnergyAlign, a dataset tailored for studying energy-oriented malicious instructions. It addresses an overlooked area of vulnerability in large language models.
- This paper explores advanced jailbreak techniques, testing their ability to bypass proprietary LLM defences against energy-based attacks, thereby revealing critical areas for security improvement in state-of-the-art models.
- This paper benchmarks existing defence strategies and proposes energy-oriented alignment as a promising approach, setting a foundational baseline for more robust, energy-efficient defences in future LLM security

**Weaknesses:**

- The paper heavily relies on output length as the primary metric for attack success (ASR), which may oversimplify the nature of energy-based vulnerabilities. Factors such as model-specific efficiency in token generation, token density, and variability in response quality are not deeply explored. Alternative or compound metrics that capture computational efficiency, memory load, or latency impacts could offer a more nuanced understanding of the resource burden and add value to the dataset.
- The paper evaluates primarily static defences, such as input filtering and pre-trained refusal criteria, without addressing adaptive, real-time detection techniques that could adjust responses based on dynamic monitoring of output behaviour. For example, adaptive rate-limiting or real-time token stream analysis could help identify and counteract excessively lengthy responses, offering a more resilient model defence approach that extends beyond pre-aligned responses.
- Although EnergyAlign is carefully crafted, it remains synthetic and may not fully capture the complexity of real-world energy-oriented attacks. Synthetic data may lack the subtle contextual variability seen in genuine user interactions, potentially limiting the dataset's ability to generalize effectively in production settings where adversaries employ context-aware or multi-layered input manipulations. Real-world data validation or augmentation could enhance the dataset’s realism and impact.
- Although various jailbreak techniques are tested, the paper’s approach may not exhaustively address all potential vectors for bypassing model restrictions. Current jailbreak techniques, such as ASCII and prefix injection, may be less representative of evolving adversarial approaches that use multi-stage prompts, obfuscation, or subtle contextual shifts. Incorporating more complex, adversarially crafted input sequences would improve robustness evaluations and provide a clearer picture of LLM susceptibility to jailbreaks.

**Questions:**

See Weakness.

---

### Note · Authors · 2024-12-01

I have read and agree with the venue's withdrawal policy on behalf of myself and my co-authors.